Promoter from Chinese hamster elongation factor-1a gene and Epstein-Barr virus terminal repeats concatemer fragment maintain stable high-level expression of recombinant proteins

Sinegubova Maria V. mvsineg@gmail.com
Orlova Nadezhda A.
Vorobiev Ivan I.
Laboratory of Mammalian Cell Bioengineering, Institute of Bioengineering, Research Center of Biotechnology of the Russian Academy of Sciences , Moscow , Russia
Gomez-Casati Diego
Electronic publication date: 2023 Oct 24
Publication date: 2023
Volume: 11
Electronic Location ID: e16287
Received 2023 Apr 14; Accepted 2023 Sep 22
Copyright: ©2023 Sinegubova et al.
Copyright year: 2023
Copyright holder: Sinegubova et al.
License: This is an open access article distributed under the terms of the Creative Commons Attribution License, which permits unrestricted use, distribution, reproduction and adaptation in any medium and for any purpose provided that it is properly attributed. For attribution, the original author(s), title, publication source (PeerJ) and either DOI or URL of the article must be cited.
License URL: https://creativecommons.org/licenses/by/4.0/

Keywords: CHO cell culture, EEF1A1 promoter, CMV promoter, Epstein-Barr virus long terminal repeat (EBVTR), High-level stable expression

Funding: The authors received no funding for this work.

==============================
Background

The Chinese hamster ovary (CHO) cell line is the main host for the high-titer production of therapeutic and diagnostic proteins in the biopharmaceutical industry. In most cases, plasmids for efficient protein expression in CHO cells are based on the cytomegalovirus (CMV) promoter. The autologous Chinese hamster eukaryotic translation elongation factor 1α (EEF1A1) promoter is a viable alternative to the CMV promoter in industrial applications. The EEF1A1 promoter and its surrounding DNA regions proved to be effective at maintaining high-level and stable expression of recombinant proteins in CHO cells. EEF1A1-based plasmids’ large size can lead to low transfection efficiency and hamper target gene amplification. We hypothesized that an efficient EEF1A1-based expression vector with a long terminal repeat fragment from the Epstein-Barr virus (EBVTR) could be truncated without affecting promoter strength or the long-term stability of target gene expression.

Methods

We made a series of deletions in the downstream flanking region of the EEF1A1 gene, and then in its upstream flanking region. The resulting plasmids, which coded for the enhanced green fluorescent protein (eGFP), were tested for the level of eGFP expression in the populations of stably transfected CHO DG44 cells and the stability of eGFP expression in the long-term culture in the absence of selection agents.

Results

It was shown that in the presence of the EBVTR fragment, the entire downstream flanking region of the EEF1A1 gene could be excluded from the plasmid vector. Shortening of the upstream flanking region of the EEF1A1 gene to a length of 2.5 kbp also had no significant effect on the level of eGFP expression or long-term stability. The EBVTR fragment significantly increased expression stability for both the CMV and EEF1A1 promoter-based plasmids, and the expression level drop during the two-month culture was more significant for both CMV promoter-based plasmids.

Conclusion

Target protein expression stability for the truncated plasmid, based on the EEF1A1 gene and EBVTR fragment, is sufficient for common biopharmaceutical applications, making these plasmid vectors a viable alternative to conventional CMV promoter-based vectors.

Introduction

Chinese hamster ovary (CHO) cells are the most common hosts used in the production of biopharmaceuticals such as monoclonal antibodies, enzymes, and hormones (Yusufi et al., 2017). The most important characteristic that determines the suitability of a producer cell line for industrial production is a high and stable level of heterologous protein expression. A clonal producer cell line should maintain a constant and specific productivity of 60 cell generations, which is the time expected to cover the entire path from the master cell bank to the end-of-production cells. Transcription rates of target genes are influenced by both cis-regulatory elements (core promoters, enhancers in the expression vector, or other regulatory elements present at the site of integration in the host genome) and trans-regulatory elements specific to the host cell (e.g., transcriptional factors) (Ho et al., 2015; Romanova & Noll, 2018; Xu et al., 2001; Lee et al., 2018; Dillon & Grosveld, 1993). Most expression vectors are based on strong promoters of viral origin (e.g., cytomegalovirus, CMV) that have been selected by evolution to achieve the maximum level of heterologous protein expression. Such vectors are expected to provide a high yield of the target protein in the culture of plasmid-transfected cells (Brown et al., 2015). CMV is usually the promoter of choice in the case of transient expression (Brown et al., 2015; Xia et al., 2006). However, CMV immediate-early promoter-driven expression shows a rapid decline during the activation of cellular antiviral mechanisms such as transcriptional silencing through DNA methylation (Yang et al., 2010; Hsu et al., 2010; Osterlehner, Simmeth & Göpfert, 2011). More stable high-level expression in cultured mammalian cells can be achieved using strong cellular promoters of housekeeping genes, such as eukaryotic translation elongation factor 1α 1 (EEF1A1) (Deer & Allison, 2004; Ebadat et al., 2017; Wang et al., 2017).

Based on the Chinese hamster EEF1A1 gene, we previously developed a p1.1 expression vector (Orlova et al., 2014), which we successfully used to obtain several cell lines producing therapeutic proteins, including human follicle-stimulating hormone (Orlova et al., 2019) and blood clotting factor VIII (Orlova et al., 2017). The plasmid p1.1 bears: the core EEF1A1 promoter, dihydrofolate reductase (DHFR) gene as a selection marker, long (about 5 kbp each) upstream flanking regions (UFR) and downstream flanking regions (DFR) of the EEF1A1 gene, and fragment of the Epstein-Barr virus terminal repeats (EBVTR) concatemer. The EBVTR element was shown to increase the rate of stably transfected colony formation (Cho & Chan, 2000). The inclusion of relatively long fragments of the Chinese hamster genome into the expression vector was based on the following idea: the sequences flanking the highly-expressed EEF1A1 gene may contain elements that positively regulate its transcription and would also work as part of the expression vector in CHO cells. The vectors based on regulatory elements from the Chinese hamster EEF1A1 gene were found to express several proteins at six to 35-fold higher levels in stable CHO cells and other mammalian cell lines, relative to the CMV promoter or human EF-1α promoter (Deer & Allison, 2004).

EF-1α-based vectors are expected to have lower transfection efficiency than CMV-based vectors due to their larger size (Xu et al., 2018; Campeau et al., 2001; Hornstein et al., 2016). We hypothesized that some of the sequences from the flanking regions of the EEF1A gene, which comprise nearly half of the p1.1 plasmid length, may not be critical for maintaining the target gene transcription rate or stability of this rate during long-term culture. At the same time, a significant decrease in plasmid size can improve both transfection efficiency and chromosome integration efficiency, as well as the rate of amplification of the target insert in the genome, which may be due to the greater number of copies of the plasmid insert in a typical amplification unit (Kim, Shin & Seo, 2021). The initial research resulted in the construction of the efficient EEF1A1-based plasmid, named pDEF38 (Deer & Allison, 2004), and was based on low-resolution restriction mapping of the EEF1A1 gene region. Four kbp-long fragments of the DNA surrounding the functional sites (the core promoter and transcription terminator) were included in the pDEF38 plasmid and found to be sufficient for both the efficient transcription rate and long-term transcription stability.

We studied the functionality of the UFR and DFR from the EEF1A1 gene with higher resolution and expected to find a minimal length for these regions, which would be sufficient to maintain stable high-level expression of the target proteins. We created a series of deletion mutants that coded for the enhanced green fluorescent protein (eGFP). Stably-transfected CHO DG44 cells were subjected to one step of methotrexate (MTX)-mediated genomic amplification and cultured in the absence of selection pressure to assess expression level stability.

Materials & Methods

Creation of p1.1-D1, p1.1-D2, and p1.1-D expression plasmids with different lengths of EEF1A1 DFR

p1.1-eGFP: The control plasmid p1.1, described in detail in Orlova et al. (2014), was deposited to Addgene as a plain vector (#162737) and with cloned fluorescent reporter proteins: p1.1-eGFP (#162738), p1.1-mCherry (#162772).

p1.1-D1-eGFP (Δ676 DFR) was obtained by restriction of the p1.1-eGFP with BstZ17I enzyme and self-ligation.

p1.1-D2 (Δ2,715 DFR) was obtained by BamHI restriction of the plasmid with the cloned EEF1A1 DFR pAL-3CH (Orlova et al., 2014) followed by self-ligation. After that, the truncated EEF1A1 DFR (Orlova et al., 2014) was cut at the XhoI and NheI sites and ligated with the pBL-ID-EBV-5CH plasmid (Orlova et al., 2014) cut at the SalI and XbaI sites. During this procedure, 2,715 bp were removed from the 4,264-bp DFR, and 1,549 bp of DFR remained. The resulting p1.1-D2 vector plasmid was inserted with the eGFP gene using AbsI and NheI, yielding p1.1-D2-eGFP.

p1.1-D (Δ3,644 DFR), the mutant with the shortest DFR, was created using plasmid pAL-3CHEF-F1 (Orlova et al., 2014), which was cut by NheI and XbaI. The DFR fragment was ligated into pBL-ID-EBV-5CH, restricted with XbaI enzyme, and the required orientation was selected using PCR. The resulting p1.1-D plasmid was inserted with the eGFP gene using AbsI and NheI, yielding p1.1-D-eGFP.

Creation of p1.1-Tr1, p1.1-Tr2, and p1.1-Tr3 expression plasmids with different lengths of EEF1A1 UFR

Three fragments of the EEF1A UFR region flanked by the EcoRI and FseI adaptor restriction sites were obtained using PCR and p1.1 as a template. The PCR-products were subcloned into the pAL2T vector and sequenced. The plasmid p1.1-D-eGFP, 8,995 bp in size, was cut by EcoRI and FseI enzymes, and the PCR-generated truncated variants were inserted in place of the 2,845-bp EcoRI-FseI UFR fragment.

p1.1-Tr1-eGFP (Δ3,644 DFR, Δ1,608 UFR): Using the primers AD-5CHD2083-EcoRIf and AD-5CHD-FseIr (Table 1), a 2,083-bp PCR product was obtained, cloned into pAL2T T-vector, sequenced with primers M13 and SQ-DEL2083-R (Table 1), and then a 2,076-bp EcoRI-FseI fragment was cut and transferred into the p1.1-D-eGFP plasmid restricted with the same enzymes. The total size of the preserved UFR (EcoRI-AbsI) was 3,309 bp.

Table 1 Primers and probes used for cloning (#1–10) and qPCR-based transgene copy number evaluation (#11–18).

#	Name	Sequence (5 ′ → 3′)	
1	AD-5CHD427-EcoRIf	CGAATTCTGAACGGAACTCCC	
2	AD-5CHD1291-EcoRIf	AGAATTCCCACATGTGGACACC	
3	AD-5CHD2083-EcoRIf	CGAATTCTGGATTCTTTGGTTACA	
4	AD-5CHD-FseIr	GGAGGCCGGCCAGAATTT	
5	SQ-DEL427-R	TGTGTCGAGATCCGGGTGTC	
6	SQ-DEL1291-R	ACTCACAGAGATCCGCCCTGCCTC	
7	SQ-DEL2083-R	CAACTCAGAGTCAGGCTTTG	
8	AD-5CHD2083-EcoRIf	CGAATTCTGGATTCTTTGGTTACA	
9	AD-5CHD2083-EcoRIf	CGAATTCTGGATTCTTTGGTTACA	
10	SQ-EBV-F	CCGCCGCTCGCCCGCCGTTG	
11	RT-eGFP-F	CAAAGACCCCAACGAGAAGC	
12	RT-eGFP-R	CTTACTTGTACAGCTCGTCCATG	
13	RT-D-F	CTGCATCGTCGCCGTGTC	
14	RT-D-R	AGTACTTGAACTCGTTCCTGAGC	
15	RT-Rab-F	GAGTCCTACGCTAATGTGAAAC	
16	RT-Rab-R	TTCCTTGGCTGTGGTGTTG	
17	VIC-eGFP	VIC-CTGCTGGAG(T-BHQ1)TCGTGACCGCCGC	
18	Cy5-DHFR	Cy5-CCAGGGTAGGTCTCCGTTCTTGCCA-(BHQ3)	

p1.1-Tr2-eGFP (Δ3,644 DFR, Δ2,400 UFR): Using the primers AD-5CHD1291-EcoRIf and AD-5CHD-FseIr, a 1,291-bp PCR product was obtained, cloned into the pAL2T vector, sequenced with primers M13 and SQ-DEL1291-R (Table 1), and then a 1,284-bp EcoRI-FseI fragment was cut and transferred into the p1.1-D-eGFP plasmid restricted with the same enzymes. The total size of the preserved UFR (EcoRI-AbsI) was 2,517 bp. The resulting p1.1-Tr2-eGFP plasmid is available from Addgene, plasmid #162782.

p1.1-Tr3-eGFP (Δ3,644 DFR, Δ3,264 UFR): Using the primers AD-5CHD427-EcoRIf and AD-5CHD-FseIr, a 427-bp PCR product was obtained, cloned into the pAL2T vector, sequenced with primers M13 and SQ-DEL427-R (Table 1), and a 420-bp EcoRI-FseI fragment was cut and transferred into the p1.1-D-eGFP plasmid restricted with the same enzymes. The total size of the preserved UFR (EcoRI - AbsI) was 1,653 bp.

Creation of expression plasmids to evaluate the functional role of EBVTR

p1.1-Tr2(-)EBVTR-eGFP was obtained by the p1.1-Tr2-eGFP restriction with EcoRI and PvuI The 959 nt fragment containing EBVTR and a part of the ampicillin resistance gene was deleted and changed to the 551 nt EcoRI-PvuI fragment of the p1.1(EBVTR-) plasmid (Orlova et al., 2014) containing no EBVTR.

pOptiVEC-eGFP was obtained on the basis of the pOptivec-circ plasmid (self-ligated pOptivec-TOPO, Thermo Fisher Scientific, Waltham, MA, USA), which was restricted withBamHI-NotI and ligated to the BamHI-NotI fragment of the pEGFP-N2 (Clontech, Mountain View, CA, USA), which contained the eGFP ORF.

pOptiVEC(+)EBVTR-eGFP was obtained by restriction of the pOptivec-circ with SpeI and PvuI enzymes and ligation with the EBVTR-containing NheI-PvuI fragment of the pBL2-ID-EBV (Orlova et al., 2014). The eGFP gene was cloned by BamHI-NotI.

The specific primers used for cloning are listed in Table 1. We used restriction endonucleases from Sibenzyme (Novosibirsk, Russia) and oligonucleotides, PCR reagents, and nucleic acid purification kits from Evrogen (Moscow, Russia). Cloning was performed in the Escherichia coli TOP10 strain (Invitrogen, Carlsbad, CA, USA), and competent cells were prepared in-house. DNA sequencing was carried out at the Interinstitutional Center for Collective Use “GENOME” IMB RAS. The genetic maps of all the plasmids are provided in Data S1.

CHO DG44 cell culture

A DHFR−/− CHO DG44 cell line (Thermo Fisher Scientific, Waltham, MA, USA) was used. The untransfected cells were cultured in 125 mL shake flasks with 30 mL of ProCHO5 medium (Lonza, Basel, Switzerland) supplemented with 4 mM glutamine, 4 mM alanyl-glutamine, and HT (in the form of 500 µM sodium hypoxanthine, 80 µM thymidine, all supplements from PanEco, Moscow, Russia) at 37 °C in a 5% CO2 incubator. The cells were passaged every 3–4 days. Cell density and viability were determined using the trypan blue exclusion method and a haemocytometer.

Transfection of CHO DG44 cells

Electroporation was performed using the Neon Transfection System (Thermo Fisher Scientific). Fifty micrograms of each plasmid were ethanol-precipitated, dried under laminar air flow, and dissolved in 50 µL of R-buffer (Thermo Fisher Scientific). The cells were split 24 h before transfection. Ten million cells were washed once in Dulbecco’s phosphate-buffered saline (DPBS), resuspended in 50 µL of R-buffer, and gently mixed with the appropriate plasmid DNA, resulting in a total volume of 100 µL. The whole volume was placed in the electroporation tip and pulsed once (1,700 V, 20 ms). The transfected cells were transferred into a 125 mL Erlenmeyer flask containing 30 mL of prewarmed ProCHO5 medium supplemented with 8 mM glutamine and HT. The transfection efficiency was assessed 48 h after transfection as the GFP fluorescence in relation to viable cell density.

Selection of stable transfectants and genomic amplification of the transgene

Selection was performed by replacing the transfection medium with ProCHO5 medium containing 200 nM MTX. The cells were passaged every 3–5 days with centrifugation (300 g, 5 min) until the cell viability was restored to 80%. One-step genomic amplification was driven by increasing MTX concentration to 2,000 nM, and the cells were passaged until the cell viability was restored to 80% (typically approximately 15 days or five passages).

Samples of stably transfected and amplified cells were collected for eGFP expression level analysis (flow cytometry and measurement of eGFP concentration in cell lysates).

Stability studies

Stability studies were conducted for 2,000 nM MTX-amplified cultures, unless stated otherwise. To assess the long-term stability of the eGFP expression level, the cells were cultured for two months (63 or 68 days) in the absence of selection pressure with 3–4 day passages using centrifugation (300 g, 5 min), seeding density 3  × 105 cells/mL. At every two or three passages, cell samples were collected for eGFP expression level analysis (flow cytometry and measurement of eGFP concentration in cell lysates).

Flow cytometry

The cells were analyzed during the exponential growth phase using a NovoCyte flow cytometer (Agilent, Santa Clara, CA, USA). Data were acquired for at least 50,000 individual cells using 488 nm excitation with a 530/30 nm bandpass filter to detect eGFP and analyzed using NovoExpress software (Agilent).

Measurement of eGFP concentration in cell lysates

The cell culture samples containing approximately 5 × 106 live cells were centrifuged at 300 g, and the pellets were washed once with PBS and stored at −20 °C for further analysis. The pellet was resuspended in 100 µL of lysis buffer containing 100 mM Tris-HCl pH 8.0, 100 mM sodium chloride, 10 mM EDTA pH 8.0, 0.5% (v/v) Triton X-100, and 1x Protease Inhibitor Cocktail (Sigma, Burlington, MA, USA); vortexed; incubated for 5 min at RT; and then centrifuged for 2 min at 13,400 rpm. The supernatant was transferred into a new tube and centrifuged for 5 min at 13,400 rpm, and the lysate was transferred to a new tube and stored at −20 °C for further analysis.

The enhanced GFP content in the lysates was evaluated by measuring fluorescence intensity at an emission wavelength of 509 nm using the Fluorat-02-Panorama spectrofluorometer (Lumex, St.Petersburg, Russia). An external calibrator, rTagGFP2 (Evrogen), was used as a fluorescence standard. Total protein content in the lysates was measured using the Bradford method with Brilliant Blue Protein Reagent (Sigma). The eGFP expression level was calculated as the eGFP concentration divided by the total protein concentration in the cell lysates.

Quantitative real-time PCR

The transgene copy number in the genomic DNA was evaluated using qPCR. Genomic DNA was isolated from 2.5 × 106 cells using the Wizard SV Genomic DNA Purification System (Promega, Madison, WI, USA) and quantified using a Qbit fluorimeter (Thermo Fisher Scientific), Qbit DNA HS kit, and 5 ng of DNA for each reaction.

For eGFP and DHFR genes, multiplex qPCR was performed using the qPCRmix-HS UDG reaction mixture (Evrogen) and two combinations of primers and fluorescent probes (Table 1), with the p1.1-eGFP plasmid as a standard. For the Rab1 gene, the qPCRmix-HS SYBR reaction mixture (Evrogen) and the pGEM-Rab1 plasmid were used.

Primers and probes used for qPCR were purchased from Lumiprobe (Moscow, Russia). The iCycler iQ thermocycler (Bio-Rad, Hercules, CA, USA) was used for amplification and detection. The cycling parameters were as follows: initial denaturation for 10 min at 95 °C and 40 cycles of amplification (10 s denaturation at 95 °C, 15 s annealing at 55 °C, 15 s elongation at 72 °C). Each sample was analyzed in triplicate. The iCycler Iq4 program was used to determine threshold cycles, PCR efficiency, and copy numbers. The weight of one CHO haploid genome was 3 pg, according to (Gregory et al., 2007).

Statistical analysis

T-tests were performed using Prism (GraphPad Software, San Diego, CA, USA), available at https://www.graphpad.com/quickcalcs/ttest1.cfm (accessed on March 18, 2023).

Results

Effects of DFR deletions

According to data from Deer & Allison (2004), the complete removal of the 4.2 kbp-long DFR of the EEF1A1 gene from the expression vector resulted in a significant (1.7- to five-fold) decrease in the target protein expression level. The full-length vector p1.1 employed by our group differed from the vector pDEF38 used in the initial study in that it contained the EBVTR genetic element, which was not expected to affect the transcriptional stability of the target gene. The DFR of the Chinese hamster EEF1A1 gene in the plasmids pDEF38 and p1.1 contained two Alu-like repeats (Haynes et al., 1981) with the predicted microRNA copies. To check their effect on target gene expression stability, we deleted one (p1.1-D1-eGFP, Δ676 bp) or both (p1.1-D2-eGFP, Δ2,715 bp), as shown in Fig. 1. The resulting plasmids were transfected into CHO DG44 cells, and the selection was performed at 200 nM MTX and stably transfected cell populations were studied in the simple batch culture for 80 days without MTX. All tested plasmids gave rise to polyclonal cell populations with similar eGFP expression levels at 200 nM MTX and demonstrated a similar slight drop in eGFP expression after 80 days of culture, less than a 20% decrease from the initial values (Fig. S1A). The copy number of the genome-integrated target genes, determined by qPCR analysis, dropped four to 10-fold on day 80 (Fig. S1B). This drop was consistent with observations made by other groups. Under similar conditions, a three-fold drop in copy number was observed for HEK293 dhfr- cells and plasmids with the DHFR marker (Lee, Baek & Lee, 2021). No correlation between copy number and specific productivity was observed (Lee, Baek & Lee, 2021).

Figure 1 Scheme of deletions for p1.1 expression plasmid in the upstream and downstream flanking regions of the EEF1A1 gene.

EEF1A1, eukaryotic translation elongation factor-1 alpha-1; UFR, upstream flanking region; TATA, TATA-box; PTS, putative transcription start; eGFP, enhanced green fluorescent protein; IRES, internal ribosome entry site; DHFR, murine dihydrofolate reductase; DFR, downstream flanking region; EBVTR, concatemer fragment of the long terminal repeat from the Epstein-Barr virus; bla prom, promoter of the ampicillin resistance gene; AmpR(bla), ampicillin resistance gene; pUC ori, bacterial replication origin; blue triangles in DFR, Alu-like repeats.

The deleted Alu-like repeats and supposed microRNA coding regions did not affect the target gene transcription level or stability of target gene expression. We observed a statistically significant 27% decrease in the amplification rate upon the deletion of the DFR. This effect is expected to be the consequence of the elimination of two Alu-like elements in the DFR that are known to contribute to the double-strand break reparation (Morales et al., 2015). The double-strand breaks occur in large numbers during MTX-driven target gene amplification (Baik, Han & Lee, 2021) and drive the amplification events. This effect of the amplification rate decrease was moderate and completely negated by the truncation of the UFR.

The deleted DFR region was considered expendable and completely removed from the plasmid vector. The resulting plasmid, p1.1-D-eGFP (Δ3,644 bp), was reduced to 9 kbp length and subjected to further deletions in the UFR.

Effects of UFR deletions

Three more plasmids (p1.1-Tr1-eGFP, p1.1-Tr2-eGFP, and p1.1-Tr3-eGFP) with deletions in the UFR were constructed and used together with two control plasmids (full-size p1.1-eGFP and p1.1-D-eGFP with the intact UFR and deleted DFR) (Fig. 1). CHO DG44 cells were transfected by electroporation, and the transfection efficiency was 20% for p1.1 and between 3% and 9% for other plasmids. Stably transfected cell populations were generated for all plasmids simultaneously in the presence of 200 nM MTX and subjected to one round of target gene amplification by increasing MTX concentration to 2,000 nM. MTX-driven genomic amplification is widely used for the generation of highly productive cell lines and is related to the increased overall genomic instability of cell lines and lower expression level stability of target genes due to unwanted chromosomal rearrangements through double-strand break repair (Baik, Han & Lee, 2021) and other effects. As shown in Fig. 2A, the eGFP expression level was nearly the same for all populations after the initial selection, but differences appeared after genomic amplification. The eGFP level was inversely correlated with the plasmid size. The eGFP expression level was the highest (4.57 ± 0.09% of total cellular protein) for the shortest plasmid (p1.1-Tr3) and was 3.25 ± 0.10% for the longest initial plasmid (p1.1). The amplification rate, calculated as the ratio of the eGFP content before and after the genomic amplification, was inversely correlated with the plasmid size for all truncated plasmids, but was higher than expected for the initial long p1.1 plasmid (Fig. 2B).

Figure 2 Enhanced green fluorescent protein (eGFP) expression level in stably transfected (200 nM MTX), gene-amplified cell cultures (2,000 nM MTX), and during long-term cultivation without MTX for EEF1A1 promoter-based plasmids.

p1.1 = full-size plasmid; p1.1-D = DFR-deleted ( Δ3,644 bp), UFR-intact; p1.1-Tr1 = DFR-deleted ( Δ3,644 bp), UFR-deleted ( Δ1,608 bp); p1.1-Tr2 = DFR-deleted ( Δ3,644 bp), UFR-deleted ( Δ2,400 bp); p1.1-Tr3 = DFR-deleted ( Δ3,644 bp), UFR-deleted ( Δ3,264 bp).(A) Level of intracellular eGFP normalized to total protein level, %. Selection pressure 200 nM MTX (light green) and 2,000 nM MTX (green). All eGFP levels data are presented as mean ±standard deviation, n = 2 (* p < 0.05, ns, not significant, unpaired t-test) if not stated otherwise. (B) eGFP expression level increase, times after one-step target gene amplification, performed by increasing MTX concentration from 200 nM to 2,000 nM. (C) Decrease in eGFP expression level during long-term cultivation for 63 days in the absence of MTX. T-test performed for p1.1-Tr2 population and p1.1-Tr3 population. All populations, except the p1.1-Tr3, have no significant difference in eGFP levels for all time points shown (two-way ANOVA). (D) Distribution of cells with different fluorescence intensity for 2,000 nM cultures during stability studies analyzed by flow cytometry (green –day 0, blue –day 63 without MTX). The numbers inside the plots indicate the percentage of cells within the gate of the 10% lowest fluorescence intensity as gated for p1.1 population (day 0/day 63). (E) Number of copies of genome-integrated plasmids measured by qPCR. Amplicons are located inside the eGFP ORF (green) and the DHFR ORF (blue). All qPCR data are presented as mean ±standard deviation, n = 3–4.

Gene-amplified cell populations were cultured in the absence of selection pressure and HT supplementation for 63 days. These culture conditions closely mimic the industrial process for producer cell lines, and the stability of the target gene expression rate for 60 cell generations is usually considered sufficient for master cell bank validation. We observed that in the absence of selection pressure, all cell populations gradually decreased eGFP expression (Fig. 2C), which coincided with the loss of a transgene inserted in the genome as measured by qPCR using primers for eGFP and DHFR ORFs (Fig. 2E). We performed long-term culture for the polyclonal populations, but not for clonal cell lines. Therefore, individual cells with a higher growth rate and lower content of recombinant protein will take over the population during the two months of culturing. Previously, we demonstrated that many clonal cell lines based on the p1.1 plasmid are stable for 60–90 days of cultivation in a non-selective medium (Orlova et al., 2017; Orlova et al., 2014; Orlova et al., 2019; Kovnir et al., 2018), so we considered the eGFP production drop for the p1.1-eGFP plasmid as a minimal possible decrease.

On day 63 of the long-term culture, there was a statistically significant productivity decrease for the p1.1-Tr3 plasmid in comparison to the p1.1-Tr2 plasmid, and the productivity decrease for the p1.1-Tr2 plasmid and all longer plasmid variants were indistinguishable from the full-size p1.1 control plasmid (Fig. 2C). The flow cytometry analysis data (Fig. 2D) were consistent with the fluorometric analysis of the cell lysates. Over time, minor subpopulations with a very low eGFP signal developed in all populations. Only the p1.1-Tr3 population also had a major subpopulation with a reduced eGFP signal, visible on the diagram as the left shoulder of the eGFP signal strength distribution peak. For the p1.1-Tr3, the highest percentage of cells (53.3%) fell into the area of fluorescence signal strength of the least productive 10% of cells (gate for the p1.1 population).

The production drop was similar for all truncated plasmids except for p1.1-Tr3. The shortest plasmid, p1.1-Tr2, had an acceptable target gene expression loss rate and was taken for further analysis.

Effect of the EBVTR fragment

The major difference between the p1.1 plasmid and the pDEF38 plasmid, developed by Deer & Allison (2004), is the presence of the 400-bp non-coding EBVTR element: the fragment of the EBV terminal repeats concatemer. EBVTR greatly increased the chromosomal integration rate of the p1.1-eGFP plasmid (Orlova et al., 2014). We hypothesized that the EBVTR element may also affect target gene expression. We removed it from the p1.1-Tr2-eGFP plasmid and compared the p1.1-Tr2-eGFP and p1.1-Tr2(-)EBV-eGFP plasmids by simultaneously preparing two stably transfected and gene-amplified cell populations, as described above. The level of eGFP expression was indistinguishable at both 200 nM and 2,000 nM MTX (Figs. 3A, 3B). The decrease in the eGFP expression level at 63 days of culture in a non-selective medium was higher for the (-) EBV plasmid, and the difference was statistically significant on day 54 (Fig. 3C). The flow cytometry data showed that at the end of the long-term culture, the pool containing the p1.1-Tr2 plasmid retained 69.6% of the initial mean fluorescence intensity versus 59.1% for the p1.1-Tr2(-)EBV plasmid (Fig. 3D). This change was visible in the histogram as the peak shift towards the lower fluorescence intensity. The percentage of the lowest expressing cells at the end of the stability run was roughly the same (41.1% versus 44.4%) for both plasmids. We observed the same transgene copy numbers for both plasmids at the end of the stability studies, but a higher eGFP level for the plasmid with EBVTR (Figs. 3C, 3E). Therefore, the EBVTR fragment was shown to affect expression level stability not by maintaining integrated transgene copies in the host genome, but by another mechanism.

Figure 3 EGFP expression level dynamics for EEF1A1 promoter-based plasmid p1.1-Tr2 with and without EBVTR genetic element.

(A) Level of intracellular eGFP normalized to total protein level, %. (B) eGFP expression level increase after one-step target gene amplification, performed by increasing MTX concentration from 200 nM to 2,000 nM. (C) Decrease in eGFP expression level during long-term cultivation for 63 days in the absence of MTX. (D) Distribution of cells with different fluorescence intensity for 2,000 nM cultures during stability studies analyzed by flow cytometry (green –day 0, blue –day 63 without MTX). (E) Number of copies of genome-integrated plasmids measured by qPCR.

Figure 4 EGFP expression level dynamics for CMV promoter-based plasmids and EEF1A1 promoter-based plasmid p1.1-Tr2.

(A) Level of intracellular eGFP normalized to total protein level, %. (B) eGFP expression level increase after one-step target gene amplification, performed by increasing MTX concentration from 200 nM to 2,000 nM. (C) Decrease in eGFP expression level during long-term cultivation for 68 days in the absence of MTX. (D) Distribution of cells with different fluorescence intensity for 2,000 nM cultures during stability studies analyzed by flow cytometry. (E) Number of copies of genome-integrated plasmids measured by qPCR.

To evaluate the effect of the EBVTR fragment, we inserted it into a commercially available pOptiVEC plasmid (Thermo Fisher Scientific) based on the CMV promoter. The resulting plasmids pOptiVEC(+)EBVTR-eGFP, pOptiVEC-eGFP, and the positive control p1.1-Tr2-eGFP plasmid (denoted below as pOV(+)EBV, pOV, and p1.1-Tr2) were simultaneously transfected into the CHO DG44 cells and tested as described above. We observed a statistically significant difference in eGFP expression level for the 2,000 nM MTX-amplified cultures: 3.62 ± 0.19% for p1.1-Tr2, 4.93 ± 0.18% for pOV(+)EBV, and 5.49 ± 0.20% for pOV. This demonstrates that the CMV-based plasmid gives a higher yield of the model protein after one-step transgene amplification (Fig. 4A). The amplification rate was also higher for smaller CMV-based plasmids: the increase in eGFP expression level was six-fold for p1.1-Tr2 (7.4 kbp), 14-fold for pOV (5.1 kbp), and 10-fold for pOV(+)EBV (5.5 kbp) (Fig. 4B). The CMV promoter-based plasmids are industry standard due to the great strength of the promoter, but they suffer from instability in their target gene expression levels. Unsurprisingly, in the case of the pOV plasmid, we observed a six-fold expression level drop after 68 days of culture in a non-selective medium, a two-fold drop in the case of the p1.1-Tr2 plasmid, and three-fold drop for the pOV(+)EBV plasmid (Fig. 4C). The flow cytometry data indicated that high-expressing cells completely disappeared in the case of the pOV plasmid but remained in the other two cases (Fig. 4D). Quantitatively, the area of 10% cells with the lowest eGFP fluorescence intensity (gate for the p1.1, day 0 in the long-term culture) on day 68 included 53.6% cells for the p1.1-Tr2, 73.9% of cells for the pOV(+)EBV, and 88.0% of cells for the pOV. According to the qPCR data (Fig. 4E), the pOV plasmid gave more genomic copies after genomic amplification than the other plasmids and most of these copies were retained during the culture in non-selective conditions. The resulting target gene expression level per gene copy was low, indicating promoter inactivation as the main reason for the productivity drop rather than transgene copy loss. The behavior of the pOV(+)EBV-based cell population was close to that of the p1.1-Tr2 population according to the flow cytometry data: highly-producing cells were mostly preserved, and the productivity drop coincided with the appearance of the low-producing subpopulation. Despite the superior stability of the pOV(+)EBV plasmid-based populations, this plasmid vector was not superior to the p1.1-Tr2 because the eGFP expression level after 68 days of culture in non-selective conditions was lower for the pOV(+)EBV than for the p1.1-Tr2.

Analysis of UFR in the orthologous human EEF1A1 gene

The UFR of the EEF1A1 gene deleted in the p1.1-Tr1, p1.1-Tr2, and p1.1-Tr3 plasmids were compared with a detailed map of the orthologous human EEF1A1 gene, compiled from the ENCODE project data (http://genome.ucsc.edu) (Kent et al., 2002). Figure 5 shows ChIP-seq-defined markers of chromatin transcriptional activity, including transcription factor binding sites (TFBS) and DNAase I hypersensitivity sites (DHS). The human EEF1A1 gene and its rodent orthologs have many homologous regions, but some regions upstream of the promoter are highly divergent. In these regions, the functional properties of hamster and human DNA can differ greatly.

Figure 5 Functional map of upstream part of human EEF1A1 gene according to ENCODE project data.

Abbreviations: EEF1A1, human EEF1A1 RNA; Layered H3K27Ac, levels of the H3K27 acetylation for various human cell lines, as measured by the ChIP-seq; DNase Clusters, clusters of hypersensitivity to DNase I treatment; 100 vert.cons., evolutionary conservation level inside vertebrates; CriGri1, Chinese hamster genome. Promoter and UFR DNA for the p1.1-Tr1; p1.1-Tr2; p1.1-Tr3 are marked by black arrows. Intact 5′-EEF1 UFR, promoter and the UFR area in the p1.1 plasmid. EEF1A1 gene transcription direction is shown by the red arrow.

The shortest version of the truncated p1.1 plasmids that we studied, p1.1-Tr3, closely matched commercially available expression vectors with human or hamster EF1 α core promoters. The truncated EEF1A1 UFR of the p1.1-Tr3 retained two complete DHS clusters out of five and one DHS fragment, two complete TFBS clusters and a cluster fragment, and one complete and one incomplete histone H3 Lys27 hyperacetylation site (H3K27ac site) out of five. Most of the functional sites identified by ENCODE for the EEF1A1 gene UFR were absent, which may explain the lack of expression stability for p1.1-Tr3 plasmid.

In the case of the p1.1-Tr2, three TFBS clusters, DHS, and the total number of H3K27a sites out of five located upstream of the EEF1A1 promoter remained intact. About 250 bp upstream of the EEF1A1 gene included in the p1.1-Tr2 were not homologous between humans and rodents. We assume that this truncated variant preserves the entire functionally significant UFR. Two more potential TFBS clusters located further upstream were removed from this plasmid variant.

The full-length EEF1A1 UFR, compared to the p1.1-Tr1, contained a possible mark of chromatin transcriptional activation - a DNA region with simultaneous increase in histone H3K27 acetylation and TFBS and DNAase I hypersensitivity. At the same time, significant homology for this DNA site was observed only between H. sapiens and M. musculus, but not between H. sapiens and R. norvegicus or between H. sapiens and C. griseus. Thus, it is likely that this region may have some equivalent in the Chinese hamster genome, but its exact position relative to the EEF1A1 promoter and its internal structure remains unknown.

Discussion

Flanking sequences together with the core EEF1A1 promoter into the expression plasmid might be useful due to the positive regulation of TFBS and the genetic elements, which prevented the heterochromatinization of the insert region. The first intron of the Chinese hamster EEF1A1 gene was 62% identical to the orthologous human intron (Wakabayashi-Ito & Nagata, 1994), and contained numerous potential TFBS (Boulikas, 1994). It was shown that both the closest UFR and the first intron of the human EEF1A1 gene are essential for its promoter activity (Wakabayashi-Ito & Nagata, 1994).

Deer & Allison (2004) reported that the 4.2 kbp EEF1A1 DFR deletion leads to a decrease in the level of protein expression (1.7- to five-fold drop, depending on the UFR length). UFR truncation was also detrimental to the promoter strength, but to a lesser extent. We found that UFR truncation to a length of 1.6 kbp (p1.1-Tr3) did not affect the absolute eGFP expression level either for the stably transfected cell population or for the gene-amplified population, but led to a more rapid decline in eGFP expression during long-term culture. This observation is partially in line with previous data (Deer & Allison, 2004). Cell cultures, described in Deer & Allison (2004), might be passaged without full selection pressure and might be equivalent to our MTX-deprived cell populations.

The p1.1 plasmid and its derivatives differed from the pDEF38 plasmid in two ways: first, we used an IRES-linked DHFR open reading frame instead of a separate selection marker gene controlled by the weak SV40 promoter and terminator, and second, we added the EBVTR element. The IRES-mediated translation of the DHFR ORF is expected to provide a very tight coupling between the target protein and selection marker expression level, making it almost impossible to uncouple the expression of the selection marker at the expense of a longer and less stable mRNA. This feature allows multistep target gene amplification, which is useful for difficult-to-express proteins such as blood clotting factor VIII (Orlova et al., 2017).

The episomal form of the Epstein-Barr virus (EBV) genome sometimes integrates into the host cell chromosomes, preferentially at the oriP and terminal repeat regions (Xu et al., 2019). The resulting provirus may activate oncogenes or disrupt tumor suppression genes, so the integration sites of the EBV in human cancer cells are extensively studied (Gao et al., 2006). Preferential integration sites are common fragile chromosome regions, and the provirus integration process is believed to be microhomology-dependent. The viral protein EBNA1 may assist in the recombination of the oriP EBV region. The recombination of the terminal repeat region and the host chromosome has not been studied in sufficient detail; the major function of the EBV terminal repeats is the circularization of the EBV genome in the nucleus through DNA recombination events. One may hypothesize that the terminal repeat fragments employ some cellular proteins for DNA recombination, so the EBVTR-containing DNA may still efficiently integrate into cellular chromosomes. The chromosome-integrated EBV genome may benefit from the possible euchromatinization activity of the flanking sequences (oriP and terminal repeats), but this topic has not been studied to date.

Current reports on the strengths of the EEF1A1 and CMV promoters vary depending on many factors, including expression cassette design, host cell line, and genomic cis-acting sequences. A number of authors report that the Chinese hamster EEF1A1 promoter is stronger than the CMV promoter, both in homologous (Ebadat et al., 2017) and heterologous (Kwok-Keung Chan et al., 2008) expression systems. In our experiments, the CMV promoter-driven eGFP expression in the amplified cell pools was on average 1.3-fold higher than the EEF1A1 promoter-driven expression, which is consistent with the results obtained in Ho et al. (2015) for stably transfected CHO K1 cells. Target gene amplification in the genome is often used to obtain highly productive cell populations and clonal cell lines, but increased productivity usually comes at the price of genetic instability and a productivity drop in the absence of MTX (Pallavicini et al., 1990; Weidle, Buckel & Wienberg, 1988; Fann et al., 2000). Target gene amplification may lead to higher binding of transcription factors to the target gene promoter region. This was shown for the CMV promoter and the CREB1 transcription factor in the CHO cells, but was not true in the case of NFκB transcription factors (Dahodwala et al., 2019). The same effect may exist for the EEF1A1 promoter studied, which may explain the disproportional increase in eGFP expression level during the gene amplification and subsequent MTX deprivation.

In this study, genetically amplified cell populations carrying the target gene under the control of the EEF1A1 promoter proved to be significantly more stable during the long-term culture without MTX compared to the cells containing the transgene under the control of the CMV promoter. These results were consistent with a previous study (Wang et al., 2017) where the episomal maintenance of the target gene was challenged by long-term culturing with or without selection pressure, and the EEF1A1 promoter-based plasmid was found to be superior to the CMV promoter-based one.

We observed the productivity drop during culturing without MTX for all polyclonal cell populations studied. This drop might be partly due to the overgrowth of the low-producing cells with shorter doubling times. The growth advantage of low-producing cells in polyclonal populations is inevitable (Barnes, Bentley & Dickson, 2003). The magnitude of this effect is similar for cell populations with similar productivity levels. Therefore, the recorded significant differences in the declining productivity rate of the cells cultured without MTX can only be explained by differences in the genetic stability of the transgene variants.

It may be expected that smaller plasmids will benefit from higher transfection efficiency and a higher genome amplification rate. We detected only a marginal and nonsignificant plasmid size effect on the target gene expression level for stably transfected cell populations. The target protein expression rate was 20% higher for the shorter p1.1-Tr2 plasmid, and this difference was statistically significant. In a previous study (Wang et al., 2018), there was no significant correlation between plasmid size and transfection efficiency, but genome amplification rate was not studied before in this way.

An increase in the transgene copy number should, in theory, be accompanied by a proportional increase in cell productivity, at least for polyclonal populations. We observed a two- to five-fold increase in copy number during transgene amplification and a similar decrease in the culture without MTX. At the same time, the productivity of both the populations after transgene amplification and populations cultivated without MTX was much higher than that of the primary stably transfected populations. This phenomenon has been previously reported (Wang et al., 2010) and can be explained as the appearance of extrachromosomal double minutes (Singer et al., 2000) during MTX-driven amplification and the subsequent loss of these genetic elements during MTX withdrawal.

We compared the UFRs of the truncated plasmids with the functional map of the human ortholog gene UFR and found that p1.1-Tr2, but not the p1.1-Tr3 plasmid, contained all three functionally active DNA regions immediately upstream of the core EEF1A1 promoter. Based on our experimental data, we concluded that these regions are sufficient both for the target gene high-level expression and maintenance of the expression stability. The truncated variant of the EEF1A1-based vector plasmid p1.1-Tr2 enabled the highest level of target gene expression in long-term culture in non-selective conditions and is suitable for creating clonal producer cell lines for the biopharmaceutical industry. The p1.1-Tr2-eGFP plasmid was deposited to Addgene (#162782).

Conclusion

We found that the EEF1A1 DFR outside the transcription termination signal and a distal part of the EEF1A1 UFR were both dispensable for the long-term stability of the target gene expression level for the EEF1A1 promoter-based plasmids. In our case, the expression plasmids also contained the EBVTR genetic element, which increased expression level stability. The combination of two vector components, the Chinese hamster EEF1A1 promoter with its 2.5 kbp-UFR and the EBVTR fragment, was effective at obtaining gene-amplified, highly stable sets of producer cells. This expression plasmid design can be used to easily generate clonal CHO cell lines that are stable enough for industrial application.

Supplemental Information

Supplemental Information 1 Enhanced green fluorescent protein (eGFP) expression level in stably transfected cell cultures (200 nM MTX) and during the long-term cultivation in non-selective conditions (without MTX) for EEF1A1 promoter-based plasmids with deletions in DFR

p1.1 = full-size plasmid; p1.1-D1 = DFR-deleted ( Δ 676 bp), UFR-intact; p1.1-D2 = DFR-deleted ( Δ 2,715 bp), UFR-intact. (A) Level of intracellular eGFP normalized to total protein level, %. Selection pressure 200 nM MTX (green–day 0, light green–day 80 without MTX). All eGFP levels data are presented as mean ±standard deviation, n = 2. (B) Number of copies of genome-integrated plasmids measured by qPCR. Amplicons are located inside the eGFP ORF (green) the DHFR ORF (blue). All qPCR data are presented as mean ±standard deviation, n = 3–4.

Click here for additional data file.

Supplemental Information 2 Genetic maps for CMV- and EEF1A1-based plasmids with deletions

Click here for additional data file.

Supplemental Information 3 EGFP fluorescence for EEF1A1-based plasmids with deletions

Click here for additional data file.

Supplemental Information 4 QPCR for EEF1A1-based plasmids with deletions

Click here for additional data file.

Supplemental Information 5 EGFP fluorescence for p1.1-Tr2 plasmids (+-EBVTR)

Click here for additional data file.

Supplemental Information 6 QPCR for p1.1-Tr2 plasmids (+-EBVTR)

Click here for additional data file.

Supplemental Information 7 EGFP fluorescence p1.1-Tr2 plasmid and pOptiVEC plasmid (+-EBVTR)

Click here for additional data file.

Supplemental Information 8 QPCR for p1.1-Tr2 plasmid and pOptiVEC plasmid (+-EBVTR)

Click here for additional data file.

We thank Sergei Kovnir and Arina Piscareva (Laboratory of Mammalian Cell Bioengineering, Research Center of Biotechnology of the Russian Academy of Sciences) for the data regarding the deletions in the EEF1A1 DFR in the p1.1 plasmid. We also thank Roman Kalinin (Laboratory of biocatalysis, Shemyakin-Ovchinnikov Institute of Bioorganic Chemistry of the Russian Academy of Sciences) for assistance in obtaining the flow cytometry data.

Additional Information and Declarations

Competing Interests

Author Contributions

Patent Disclosures

Data Availability

Maria V. Sinegubova declares no competing interests. Nadezhda A. Orlova and Ivan I. Vorobiev are inventors of the patent RU2488633, which covers the use of the p1.1 plasmid; derivatives of this plasmid are used in current research. The existence of the patent does not alter our adherence to PeerJ policies on sharing data and materials.

Maria V. Sinegubova performed the experiments, analyzed the data, prepared figures and/or tables, authored or reviewed drafts of the article, and approved the final draft.

Nadezhda A. Orlova conceived and designed the experiments, performed the experiments, authored or reviewed drafts of the article, and approved the final draft.

Ivan I. Vorobiev conceived and designed the experiments, analyzed the data, prepared figures and/or tables, authored or reviewed drafts of the article, and approved the final draft.

The following patent dependencies were disclosed by the authors:

Patent RU2488633 ”Expression Plasmid Vector for Heterologous Expression of Recombinant Proteins, High-Frequency Integration, and Enhanced Amplification of the Expression Cassette in Mammalian Cells, Bicistronic mRNA, Method for Obtaining Stable Recombinant Protein-Producing Cell-Lines Using the Vector, and Method for Recombinant Proteins Production”.

The following information was supplied regarding data availability:

The raw measurements (eGFP fluorescence and qPCR) are available in the Supplementary Files.

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
