# Peer review of "Promoter from Chinese hamster elongation factor-1a gene and Epstein-Barr virus terminal repeats concatemer fragment maintain stable high-level expression of recombinant proteins"

_PeerJ, doi:10.7717/peerj.16287_

## Round 0.1 · original submission · Major Revisions

The manuscript describes the development of a new vector that shows advantages over previously used vectors. I suggest the authors take into account the criticisms of the reviewers

Reviewer 1 ·

Basic reporting

There is a need to improve english for clarity and understanding.
The authors should use "Culture" instead of "cultivation".

The authors should share the DNA sequences of the plasmids constructed.

Combine Table 1 and Table 2 into one Table.

In general, make the formatting look better.

Experimental design

The manuscript falls within Aims and Scope of the journal.

The research question is well defined, relevant and meaningful.

The experiments are well done and methods section is also well described.

The results section needs to more succinct and better organized.

There is too much text under the title "Analysis of UFR in orthologous human EEF1A1 gene"
It also seems that the results section has too much discussion. Please keep the results section focussed on the manuscript's findings.

Validity of the findings

The underlying data is provided and looks fine.

Conclusions need to stated more concisely and clearly.

Additional comments

The authors have provided almost 3 pages of discussion for results that be summarized in 1 page. Please make it concise. It goes against clarity and readability.

Reviewer 2 ·

Basic reporting

The vector development for improving the recombinant protein production in CHO cell lines as delineaated in this manuscript is definitely an upgrade to the existing expression vectors and could have a potential to be upscaled to an industry level in future. Most of their experiments are straight forward and explain the vector construction process.

Experimental design

Experimental design is clearly elaborated. The few concerns in them has to be addressed before publishing. Please refer to the annotated manuscript or my commentary.

Validity of the findings

Designing an expression vector for recombinant protein production is clearly a novelty with the fact that the authors exploited the LTR of EBVTR in improving the yield. They are convincing. The one another practical fact that could help would be submitting the plasmid to Addgene and make it available to rest of the research fraternity.

Additional comments

1. Why the background/abstract is big? Is there a word count to limit that?
2. Few of the statements were confusing. Try to rephrase or correct the language.
3. The annotated manuscript highlights the sentence structure that needs to be modified.
4. Line 79. What do they mean by "supposed"
5. Line 222. How was the stability assessed by fluorescence?
6. What is the control used in figure1?
7. How are the expression compared to other commercially available vectors?
8. Line 308. Why the day 63 was chosen?
9. Figure 2E is not cited in the main text. Other citations have to be checked to maintain in the chronological order.
10. Line 353. Copies were more but less expression (Figure 3). How is it possible? Does this mean there is a host machinery affecting the protein expression?
11. Overall language has to be corrected.

Annotated reviews are not available for download in order to protect the identity of reviewers who chose to remain anonymous.

---

## Round 0.2 · Major Revisions

The authors answered all the reviewers' suggestions and the manuscript has been improved considerably, however, the English in the text still needs to be further improved. I suggest that the manuscript be proofread by a fluent English speaker or a language editing service.

In addition, % (v/v) should be added (triton concentration, line 471)

---

## Round 0.3 · accepted · Accept

The authors reviewed and corrected the reviewers' suggestions.

Reviewer 1 ·

Basic reporting

No Comment

Experimental design

No comment

Validity of the findings

No comment

Additional comments

The authors have improved there manuscript after the revision and present a professional manuscript, that is ready to be published in PeerJ.